# Imidazole-Thiosemicarbazide Derivatives as Potent Anti-*Mycobacterium tuberculosis* Compounds with Antibiofilm Activity

**DOI:** 10.3390/cells10123476

**Published:** 2021-12-09

**Authors:** Adrian Bekier, Malwina Kawka, Jakub Lach, Jarosław Dziadek, Agata Paneth, Justyna Gatkowska, Katarzyna Dzitko, Bożena Dziadek

**Affiliations:** 1Department of Molecular Microbiology, Faculty of Biology and Environmental Protection, University of Lodz, 90-237 Lodz, Poland; adrian.bekier@biol.uni.lodz.pl (A.B.); malwina.kawka@biol.uni.lodz.pl (M.K.); jakub.lach@biol.uni.lodz.pl (J.L.); justyna.gatkowska@biol.uni.lodz.pl (J.G.); 2Biobank Lab, Department of Molecular Biophysics, Faculty of Biology and Environmental Protection, University of Lodz, 90-237 Lodz, Poland; 3Laboratory of Genetics and Physiology of Mycobacterium, Institute of Medical Biology, Polish Academy of Sciences, 93-232 Lodz, Poland; jdziadek@cbm.pan.pl; 4Department of Organic Chemistry, Faculty of Pharmacy, Medical University of Lublin, 20-093 Lublin, Poland; agatapaneth@umlub.pl

**Keywords:** thiosemicarbazide, PE-PGRS, *Mtb* biofilm

## Abstract

*Mycobacterium tuberculosis* (*Mtb*) is an intracellular pathogenic bacterium and the causative agent of tuberculosis. This disease is one of the most ancient and deadliest bacterial infections, as it poses major health, social and economic challenges at a global level, primarily in low- and middle-income countries. The lack of an effective vaccine, the long and expensive drug therapy, and the rapid spread of drug-resistant strains of *Mtb* have led to the re-emergence of tuberculosis as a global pandemic. Here, we assessed the in vitro activity of new imidazole-thiosemicarbazide derivatives (ITDs) against *Mtb* infection and their effects on mycobacterial biofilm formation. Cytotoxicity studies of the new compounds in cell lines and human monocyte-derived macrophages (MDMs) were performed. The anti-*Mtb* activity of ITDs was evaluated by determining minimal inhibitory concentrations of resazurin, time-kill curves, bacterial intracellular growth and the effect on biofilm formation. Mutation frequency and whole-genome sequencing of mutants that were resistant to ITDs were performed. The antimycobacterial potential of ITDs with the ability to penetrate *Mtb*-infected human macrophages and significantly inhibit the intracellular growth of tubercle bacilli and suppress *Mtb* biofilm formation was observed.

## 1. Introduction

Tuberculosis (TB), which is caused by the intracellular bacterial pathogen *Mycobacterium tuberculosis* (*Mtb*), is a severe disease primarily in the respiratory tract. Unfortunately, despite many years of efforts in the field of microbiological and clinical research, *Mtb* infection is still a serious problem, second in terms of mortality and infectious disease worldwide. As a microorganism characterized by high evolutionary success, *Mtb* has spread worldwide, infecting 30% of the human population. According to WHO data, approximately 10 million new cases of tuberculosis are diagnosed annually, and approximately 2 million people die because of this disease [1]. One of the reasons for the worsening epidemiological situation of TB in the world is the increasing drug resistance of the *Mtb* strains, classified as multidrug resistant, extensively drug resistant, and totally drug resistant [2,3]. Due to the increasingly limited effectiveness of TB therapy, its time-consuming nature, and the requirement for simultaneous treatment with several drugs exerting many side effects on patients, the development of new strategies to successfully cure TB is one of the most urgent challenges in modern microbiology. The general approach for developing effective anti-TB drugs is based on the modification of the currently used antibiotics as TB treatments to improve the pharmacodynamics of newly designed drugs and to enable the use of new compounds or alternative therapeutics, such as host-directed therapies, vaccines, or bacteriophages. The synthesis and identification of novel compounds with new mechanisms of antimycobacterial activity constitute a potentially valuable and innovative approach that might succeed in developing modern drugs [4,5].

Thiosemicarbazides are biologically active, noncytotoxic organic compounds with proven antitumor, antiviral, antifungal, and antioxidant activities. Furthermore, the strong antibacterial potential of various thiosemicarbazide derivatives has also been documented [6,7,8,9,10]. Although the in vitro activity of imidazolo-thiosemicarbazide against the virulent *Mtb* H37Rv strain has not previously been described, other than in the present study, many derivatives containing thiosemicarbazide moieties have been synthesized and evaluated for their antibacterial activity.

As part of global efforts to develop an efficient antituberculosis drug, our group is focused on the development of new thiosemicarbazide-based compounds as potential anti-*Mtb* agents. Because of the importance of the thiosemicarbazide moiety, we investigated the anti-*Mtb* activity of new imidazole-thiosemicarbazide derivatives (ITDs). Among the 30 studied compounds, only three exhibited strong inhibition of *Mtb* culture growth at noncytotoxic concentrations. Furthermore, one of the ITDs, ITD-**13**, inhibited the intracellular growth of *Mtb* in human macrophages. Additionally, the effect of ITD-**13** on mature mycobacterial biofilms and their formation was studied. We also analyzed the frequency of mutations by performing a fluctuation assay and conducted next-generation sequencing of *Mtb* mutants that were resistant to high concentrations of the tested chemical. According to whole-genome sequencing of ITD-**13**-resistant *Mtb* mutants, we identified the accumulation of mutations in the gene encoding PE_PGRS4.

## 2. Materials and Methods

### 2.1. Compounds

The imidazole-thiosemicarbazides **1**–**30** were prepared under previously reported procedure [11,12] as follows: a solution of 4-methylimidazole-5-carbohydrazide (0.01 mol, CBR02129, Sigma, St. Louis, MO, USA) and appropriate isothiocyanate (0.01 mol, 253731, Sigma) in anhydrous ethanol (25 mL, 396480111, POCH, Gliwice, Poland ) was heated under reflux for 10–30 min. After cooling, the solid formed was filtered off, dried and crystallized from ethanol. Molecular weight and physicochemical characterization of all compounds are presented in our previous publications [11,12]. Stock solutions (50 mg/mL) of 30 ITDs (Figure 1) were prepared in dimethyl sulfoxide (D4540, DMSO, Sigma–Aldrich, St. Louis, MO, USA), with a 1% maximum final concentration.

### 2.2. Bacterial Cultures and Cell Cultivation

The *Mycobacterium tuberculosis* H37Rv strain (ATCC^®^ 25618™, ATCC, Manassas, VA, USA) was grown as previously reported [13]. Briefly, *Mtb* was cultured in 7H9 medium (Difco, Baltimore, MD, USA) supplemented with BBL™ Middlebrook 10% Oleic Albumin Dextrose Catalase (OADC) (Difco, Baltimore, MD, USA)Enrichment and 0.05% Tween 80 (P4780, Sigma) at 37 °C for 4–6 days until the bacterial cell suspension reached an optical density at 600 nm (OD_600_) of 1.0. Additionally, growth was monitored by measuring the OD_600_ and performing CFU analyses. Preliminary experiments revealed that an OD_600_ = 1 corresponds to 1 × 10^8^
*Mtb* cells/mL. L929 cells (ATCC^®^ CCL-1™) were maintained as described in the ATCC product sheet.

### 2.3. Cytotoxicity Assay

Cytotoxicity assays were performed precisely according to international standards (ISO 10993-5:2009(E)) using L929 cells and the MTT assay. Additionally, an evaluation of the cytotoxicity of ITDs toward human monocyte-derived macrophages was performed, but in this case, the cells were treated with the derivatives for 48 h. The CC_30_ was defined as the lowest concentration of compound that led to a 30% reduction in cell viability. DMSO was used as a positive control, and each experiment was performed in triplicate.

### 2.4. Resazurin Microtiter Assay

The resazurin microtiter assay (REMA) plate method was used, as previously described [14,15]. Briefly, serial twofold dilutions of each drug were prepared directly in a sterile flat-bottom 96-well plate containing 100 μL/well of 7H9-S broth (7H9 medium additionally supplemented with 0.1% casein hydrolysate (22090, Sigma)), and the same volume of H37Rv inoculum was added. A growth control and a sterile control were also included. The plates were sealed with parafilm and incubated at 37 °C for 7 days. Next, 30 µL of a 0.02% resazurin (R7017, Sigma) solution were added to each well, and the plates were incubated for an additional 24 h. Changes in the sample color from blue to pink indicated bacterial growth. The minimal inhibitory concentration (MIC) was determined as the lowest concentration of drug that prevented the color change from blue to pink. All experiments were performed in triplicate.

### 2.5. Anti-Mtb Activity of ITDs and Time–Kill Kinetics Assay

The concentration- and time-dependent killing properties of ITDs against *Mtb* were determined using a previously described method, with some modifications [16]. Briefly, H37Rv cultures (log phase) were exposed to ITDs at two fold increasing concentrations for 4 days at 37 °C. The ITD concentrations ranged from 15.625 to 125 μg/mL. The tested concentrations were based on the MIC values of the individual derivatives. On days two and four of drug exposure, samples were collected, centrifuged at 4000× *g*, serially diluted and subcultured in 7H10 medium (Difco). CFUs were counted after incubating the plates at 37 °C for 3 to 4 weeks, and viability was reported as logCFU/mL. The lower limit of detection (LLD) was 5 CFU/mL (log 0.7). Time–kill curves and concentration–effect curves were generated, and then MIC_50_ and MIC_90_ were calculated. All experiments were performed in triplicate.

### 2.6. Preparation of Human Monocyte-Derived Macrophages (MDMs)

Human MDMs were differentiated from blood monocytes isolated from anonymized, commercially available buffy coats from healthy human blood donors (Regional Blood Donation Station, Lodz, Poland) using a double density gradient technique employing Histopaque-1077 (10771, Sigma) and 46% iso-osmotic Percoll™ PLUS (17544502, Cytiva Sweden AB, Uppsala, Sweden). The detailed methodology was based on a reported protocol [17] with slight modifications described in previous studies [13,18].

### 2.7. Intracellular Anti-Mtb Activity of ITD-**13**

The infection of human monocyte-derived macrophages with *Mtb* and evaluation of the bactericidal effect of ITD-**13** on intracellularly growing tubercle bacilli were performed as previously described [13]. Briefly, 24 h before the *Mtb* infection, cultures of MDMs were extensively washed three times to remove any nonadherent cells: once with Iscove’s modified Dulbecco´s medium (IMDM, 04-20150, Cytogen, GmbH, Greven, Germany) supplemented with 0.5% human serum (H5667, Sigma) and 1% P/S (P4333, Sigma) and twice with the same medium lacking antibiotics. Next, 1 mL of IMDM supplemented with only 10% human serum was added to each well, and the macrophages were incubated overnight. The next day, the cells were infected with *Mtb* (5 × 10^7^/mL) prepared in culture medium without antibiotics (MOI of 1:10). After two hours of incubation, the extracellularly located, nonadherent bacterial cells were removed by three washes with medium lacking antibiotics. One milliliter of medium containing 1 mg/mL gentamicin (G1522, Sigma) was added to eliminate nonphagocytosed mycobacteria, and the infected macrophages were incubated for an additional 1 h. Then, the cells were washed three times with medium lacking antibiotics. Next, culture medium lacking antibiotics but containing ITD-**13** was added to the cultures of infected macrophages. Based on the experiments assessing the mycobactericidal effect of the tested compounds, only ITD-**13**, which exhibited antimycobacterial activity at a concentration of 13.48 µg/mL and the lowest cytotoxicity, was selected for this assay. The chosen derivative was applied at the MIC_50_ and MIC_90_, namely, 13.48 and 32.60 µg/mL, respectively. Each concentration was tested in triplicate. *Mtb*-infected macrophage cultures incubated with medium lacking antibiotics served as controls. Both infected MDMs treated with ITD-**13** and the control cells were incubated at 37 °C for 48 h in a humidified atmosphere containing 10% CO_2_. Next, the macrophages were lysed with 1 mL of 0.1% SDS (71736, Sigma), and the cell lysates were serially diluted and plated onto 7H10 medium. CFUs were counted after an incubation of the plates at 37 °C for 3 to 4 weeks, and viability was reported in log CFU/mL. The LLD was 5 CFU/mL (log 0.7).

### 2.8. Mycobacterial Biofilm Formation

The effect of ITD-**13** on biofilm formation was examined for the *Mtb* H37Rv strain by performing biofilm assays in the absence and presence of the compound (MIC_50_ and MIC_90_). Growth of *Mtb* biofilms was determined in Sauton’s medium [0.5 g of KH_2_PO_4_, 0.5 g of MgSO_4_ (M7506, Sigma), 4 g of L-asparagine (A4159, Sigma), 2 g of citric acid (C2404, Sigma), 0.05 g of ferric ammonium citrate (F5879, Sigma), 60 mL of glycerol (G2025, Sigma), and 900 mL of deionized water, pH 7.0 in 24-well plates, as previously described with some modifications [19]. Immediately before the experiment, sterile ZnSO_4_ (Z0251, Sigma) was added to a final concentration of 0.1% *w/v*. Briefly, H37Rv was cultured to OD_600_ = 1 in 7H9 medium supplemented with 10% OADC and 0.05% Tween 80. Next, the inoculum was added at a ratio of 1:100 *v/v* to Sauton’s medium, and then the obtained mixture was dispensed into each well of a 24-well plate (2.5 mL/well). The plate was covered with a lid, carefully wrapped with parafilm, and further incubated undisturbed in a humidified microbiological incubator at 37 °C for 5 weeks.

#### 2.8.1. Effect of ITD-**13** on Mycobacterial Biofilm Formation

In our study, we performed two assays using mycobacterial biofilms. In the first assay, the effect of our compound on biofilm formation was examined. Selected concentrations of the chosen derivative were added to the diluted inoculum of *Mtb* in Sauton’s medium. Then, 2.5 mL of the mixture were dispensed into each well. The plates were covered with lids and carefully wrapped with parafilm. The plates prepared in this manner were incubated undisturbed in a humidified incubator at 37 °C for 5 weeks. After the incubation, biofilms were photographed. Then, the remaining medium under the biofilms was removed, and 2.5 mL of Sauton’s medium supplemented with 0.1% casein hydrolysate were added. Next, 375 µL of a 0.02% resazurin solution were added to each well, and plates were further incubated for 90 min. A growth control and a sterile control were also included. After the incubation, 200 µL of each sample were transferred to a 96-well plate, and the amount of resorufin produced was determined by measuring fluorescence (excitation: 560 nm, emission: 590 nm) using a SpectraMax^®^ i3 multimode microplate reader (Molecular Devices, San Jose, CA, USA). The sample to which the compound was not added was set to 100% viability. The results are reported as a percentage of viability compared to untreated mycobacterial biofilms.

#### 2.8.2. Effect of ITD-**13** on Mature Mycobacterial Biofilms

After 5 weeks of incubation, the remaining medium under the mature *Mtb* biofilms was removed, and 2.5 mL of Sauton’s medium containing 0.1% casein hydrolysate and supplemented with or without the compound were added. The plates were sealed with parafilm and incubated at 37 °C for 48 h. Next, 375 µL of 0.02% resazurin solution were added to each well, and the plates were further incubated for 90 min. Growth and sterile controls were also included. After the incubation, 200 µL of each sample were transferred to a 96-well plate, and the amount of resorufin produced was determined by measuring fluorescence (excitation: 560 nm, emission: 590 nm) using a SpectraMax^®^ i3 multimode microplate reader. The sample treated without the compound was set to 100% viability. The results are presented as a percentage of viability compared to untreated *Mtb* biofilms.

#### 2.8.3. Confocal Microscopy Imaging of Biofilm Structures

The *Mtb* biofilm structure was visualized using a Leica TCS SP8 confocal microscope equipped with Leica LAS AF Lite Software for image processing (Leica, Wetzlar, Germany). After 5 weeks of incubation, the formed biofilm was transferred to a glass microscope coverslip and fixed with 3.7% formaldehyde (252549, Sigma). For visualization of the biofilm structure, the preparations were stained with BacLight™ Green Bacterial Stain (B35000, Thermo Fisher Scientific, Waltham, MA, USA) according to the manufacturer’s instructions. Phenolic acridine orange fluorescent live-dead staining was employed to analyze the viability of biofilm-forming mycobacteria, as previously described [20]. Confocal microscopy imaging was performed in the Laboratory of Microscopic Imaging and Specialized Biological Techniques (Faculty of Biology and Environmental Protection, University of Lodz).

### 2.9. Fluctuation Assay

The fluctuation analysis was performed as previously described [21] with modifications. The culture of *Mtb* was grown at 37 °C until reaching an OD_600_ of 1.0, and approximately 3 × 10^5^ bacterial cells were inoculated into 120 mL of 7H9 supplemented with 10% OADC, 0.0005% Tween 80, and 0.005% glycerol to determine the frequency. The culture was grown at 37 °C with shaking for 11 to 14 days until reaching an OD of 1.0. Once at an OD of 1.0, the culture was divided into 20 cultures of 4 mL each, which were transferred to 15 mL conical tubes and centrifuged at 4000× *g* for 10 min at 4 °C. Cultures were then resuspended in 250–500 μL of 7H9/OADC/Tween/glycerol, spotted onto 7H10/OADC/Tween/glycerol plates supplemented with 5 μg/mL rifampicin (RMP, R3501, Sigma), 1 μg/mL isoniazid (INH, I3377, Sigma) and ITD-**13** at concentrations of 5 × MIC_50_ (67.4 µg/mL) or 5 × MIC_90_ (163 µg/mL) and subsequently incubated at 37 °C for 28 days.

The drug resistance rate was determined by calculating “m” (the estimated number of mutations per culture) based on the number of mutants (*r*) observed on each plate using the Ma, Sarkar, Sandri (MSS) method described previously [22]. The division of *m* by *Nt*, the number of cells plated for each culture, indicates the estimated drug resistance rate.

### 2.10. Selection of Mutants and Next-Generation Sequencing

Mutants resistant to ITD-**13** were selected on 7H10/OADC solid medium supplemented with 3 × MIC_90_ (97.8 µg/mL) and 5 × MIC_90_ (163 µg/mL) of the compound. The *Mtb* culture was (or was not) preincubated with the subinhibitory concentration of ITD-**13** (0.1 × MIC_90_) for three passages before plating.

Next-generation sequencing was performed as previously described [23]. Briefly, 1 ng of DNA isolated from the wild-type *Mtb* strain and six mutants was used to prepare paired-end libraries according to the protocol provided with the Vazyme TruePrep DNA Library Prep Kit V2 for the Illumina preparation (Vazyme Biotech Co., Ltd., Nanjing, China). Whole-genome sequencing was performed on a NextSeq 500 platform at a read length of 2 × 150 bp (300 cycles). The *Mtb* H37Rv reference sequence (NC_000926) was used to align the sequencing reads after quality check (FastQC v.0.11.8) and adapter trimming steps performed with trim_galore v.0.6.4 (http://www.bioinformatics.babra-ham.ac.uk/projects/trim_galore/, accessed on 19 October 2021. SNP variants were identified and filtered using the breseq v.0.35.4 on default parameters (https://barricklab.org/twiki/bin/view/Lab/ToolsBacterialGenomeResequencing, accessed on 19 October 2021. The sequencing data for all strains are available as the bioproject at the NCBI (National Center for Biotechnology Information) database under accession number PRJNA745506.

### 2.11. Statistical Analysis

Statistical analyses were performed and graphs were produced using GraphPad Prism version 9.0.0 for macOS (GraphPad Software, San Diego, CA, USA). The Mann–Whitney U-test was used to compare the bactericidal activity of ITD-**13** against intracellularly located *Mtb*. The bactericidal activity of imidazole-thiosemicarbazide derivatives, biofilm assays and fluctuation analysis were compared using two-way ANOVA followed by Dunnett’s multiple comparisons test. The Mann–Whitney U-test was used to compare the bactericidal and intracellular activities of ITD-**13**. The results were considered statistically significant at *p* < 0.05.

## 3. Results and Discussion

We previously evaluated the activity of imidazole-thiosemicarbazide derivatives against the parasite *Toxoplasma gondii*. The cytotoxicity of the compounds used in this study was also evaluated toward the mouse fibroblast (L929) cell line using the MTT assay, and the CC_30_ values were calculated. Cytotoxic effects were not observed after treatment with any compound up to a concentration of 125 µg/mL [11,12]. Table 1 shows the MICs of the tested imidazole-thiosemicarbazide derivatives against *Mtb* H37Rv obtained using REMA. Two compounds, ITD-**20** and ITD-**30**, possessed promising MIC values (31.25 µg/mL). However, the most potent compound was ITD-**13,** with the estimated lowest minimal inhibitory concentration against *Mtb* growth of 15.62 μg/mL. Notably, only one of 30 tested compounds (ITD-**14**) with an aliphatic chain instead of an N4 phenyl ring was not active against *Mtb*. Similar observations were reported in our previous study with another intracellular pathogen, *T. gondii* [12]. This result confirms our initial assumption regarding the favorable effect of the presence of the N4 phenyl ring in the derivative structure on inhibiting intracellular microorganisms.

The three most effective imidazole-thiosemicarbazide derivatives (ITD-**13**, ITD-**20,** and ITD-**30**) against *Mtb* selected by REMA were used for further study. The concentration- and time-dependent anti-*Mtb* activities of these ITDs are shown in Figure 2.

In the static in vitro assay, all tested ITDs showed significant concentration-dependent antimycobacterial activity (*p* < 0.05) at concentrations ranging from 31.25 to 125 µg/mL, and the results were compared using two-way ANOVA followed by Dunnett’s multiple comparisons test. ITD-**13** and ITD-**30** showed mycobactericidal activity at concentrations ≥31.25 µg/mL and 62.5 µg/mL, respectively. Additionally, the presence of either of the compounds, ITD-**13** or ITD-**30**, at concentrations of ≥125 µg/mL led to the almost complete (99.9%) elimination of the bacilli after 96 h of culture. In turn, **ITD-20** inhibited mycobacterial growth at concentrations ranging from 31.25 to 62.50 µg/mL, but it did not exert a mycobactericidal effect at concentrations ≥125 µg/mL.

The concentration–effect relationships after 48 hours of *Mtb* incubation with ITD-**13**, ITD-**20** and ITD-**30** are presented in Figure 3. The concentration–effect curves were the basis for MIC_50_ and MIC_90_ calculations. The lowest minimal inhibitory concentrations are shown for ITD-**13,** with an MIC_50_ value of 13.48 µg/mL and an MIC_90_ value of 32.60 µg/mL. The MIC_50_ and MIC_90_ values calculated for the other two compounds correspond to 18.62 µg/mL and 67.61 µg/mL for ITD-**30** and 24.55 µg/mL and 229.09 µg/mL for ITD-**20**, respectively.

Two studies have described the antibacterial potential of thiosemicarbazides against the avirulent *Mtb* H37Ra strain. By modifying the thiosemicarbazide core, researchers have shown that these compounds possess potential antimicrobial activity. Flieger et al. [24] tested the antimycobacterial activity of thiosemicarbazides and cyclization products of these compounds, such as 1,2,4-triazoles. Some compounds inhibited the growth of the *Mtb* H37Ra strain at the same or lower concentrations than the MIC of streptomycin (MIC = 512 µg/mL). The MIC values of both of the best working drugs were 512 µg/mL for Compound 19B and 256 µg/mL for Compound 21B. The lack of a substituent at the C5 position of the thiazole ring of the tested derivatives is associated with their antibacterial activity [24]. Recently, another in vitro study reported the activity of thiosemicarbazide derivatives against *Mycobacterium bovis*. Sardari et al. [9] condensed 4-phenylthiosemicarbazides or thiosemicarbazides with aldehydes or ketones and synthesized new compounds with antimycobacterial activity. The authors tested compounds against *M. bovis,* which has a high genetic similarity to pathogenic *Mtb*. Using the resazurin test, they showed a promising antimycobacterial effect of some compounds with MIC values approximately two times lower (0.39 µg/mL) than that of ethambutol (0.75 µg/mL) [9]. Furthermore, thiacetazone is one of the oldest and cheapest second-line drugs used for tuberculosis treatment. Unfortunately, this drug shows only bacteriostatic activity, and *Mtb* bacilli easily develop resistance during tuberculosis therapy. Additionally, the use of thiacetazone along with ethionamide promotes cross-resistance to ethionamide. Finally, thiacetazone is excluded from use in patients with HIV because it frequently causes Stevens–Johnson syndrome [25]. Newer agents bearing thiosemicarbazide motifs are being investigated in clinical trials for treating tuberculosis, such as SRI-286 [26] and KBF-611 [27], to counter these effects. The abovementioned reports on the anti-*Mtb* activity of thiosemicarbazide derivatives and the increasing drug resistance of *Mtb* against currently used first- and second-line drugs inspired research to obtain effective compounds with antimycobacterial activity belonging to this chemical group.

As an intracellular pathogen that has achieved evolutionary success, tubercle bacilli grow inside the alveolar macrophages of the host and are protected from drugs that are unable to penetrate inside human phagocytes. Therefore, an important feature of modern antituberculosis drugs is not only their mycobactericidal effectiveness but also their ability to penetrate the intracellular milieu of *Mtb*-infected cells, constituting potential niches for intracellular multiplication of these bacteria. A cytotoxicity screen against MDMs was performed to ensure that selected ITDs targeted the tubercle bacilli exclusively and did not affect the host target phagocytes, namely, macrophages. Cytotoxicity, which was reported as CC_30_, was defined as the concentration of the tested compound that caused the destruction of ≥30% of macrophages after 48 h of incubation. In Figure 4, the results are presented as the percentage of viable cells ± standard deviation. Based on the experimental data, only ITD-**13** exhibited a favorable CC_30_ cytotoxicity index at a concentration of 68.08 µg/mL, which exceeded the concentrations calculated for its MIC_50_ and MIC_90_ values (Figure 4). Although the CC_30_ concentrations estimated for the other two compounds were higher than those for their MIC_50_, namely, 33.04 µg/mL (ITD-**20**) and 51.02 µg/mL (ITD-**30**), they were two and four times lower than the corresponding concentrations of the MIC_90_ values.

Finally, we evaluated the activity of the most effective antimycobacterial compound ITD-**13** against intracellularly located *Mtb* in human macrophages. The compound was applied at the minimal concentrations that inhibited the growth of microbiologically propagated *Mtb* by 50% and 90%, namely, 13.48 µg/mL and 32.60 µg/mL, respectively. The *Mtb*-infected MDMs were incubated with ITD-**13** for 48 h. The number of intracellularly located live bacteria was calculated by measuring CFUs, and the viability was reported in logCFU/mL (Figure 5). The results showed the ability of ITD-**13** to penetrate macrophages, suggesting that they are especially attractive for the elimination of intracellularly located tubercle bacilli. We observed a significant reduction in the number of viable bacilli isolated from *Mtb*-infected MDMs cultured in medium supplemented with ITD-**13** by 35% (*p* = 0.025) and 60% (*p* < 0.0001) after incubations with concentrations of the compound corresponding to MIC_50_ and MIC_90_, respectively.

The first step in the confrontation of *Mtb* with immune response mechanisms after passing through natural mechanical barriers [28,29,30] is the recognition of the pathogen by cells involved in nonspecific immune mechanisms, including lung-resident alveolar macrophages [31]. An important element of innate immune mechanisms in the course of primary infection with *Mtb*, which is conditioned by the functional activity of alveolar macrophages, is phagocytosis, which enables pathogen elimination. Due to the diverse phenotype of pulmonary macrophages, including both classically (M1) and alternatively (M2) activated macrophages, these cells are characterized, on the one hand, by high phagocytic activity and, on the other hand, by low bactericidal activity, thus constituting an adequate niche for intracellular multiplication of *Mtb* [32,33,34]. The ability of compounds to penetrate human alveolar macrophages is undoubtedly essential for the treatment of pulmonary infections caused by intracellular pathogens. Thus, an essential direction of research aiming to identify new compounds with anti-*Mtb* activity, in addition to determining the mycobactericidal activity of compounds in medium cultures, is also to determine their ability to penetrate inside phagocytes and to successfully eliminate the intracellularly located bacilli. In this paper, among the 30 analyzed imidazole-thiosemicarbazide derivatives, three potential compounds (ITD-**13**, ITD-**20**, and ITD-**30**) inhibiting the growth of *Mtb* were reported. Furthermore, one of them, ITD-13, which was characterized by low cytotoxicity toward monocyte-derived human macrophages, also exhibited an ability to significantly inhibit the intracellular growth of *Mtb*. Therefore, imidazole-thiosemicarbazide derivatives might be particularly interesting in the context of further analyses to determine their potential use in TB therapy.

A similar anti-*Mtb* effect on intracellular bacteria was observed for compounds in the study by Korycka-Machała et al., namely, carbohydrate and 1H-benzo[d]imidazole derivatives [13,35]. The number of intracellularly growing *Mtb* in the presence of the MIC_50_ of tested Compound 16A (25.53 µg/mL) was decreased by approximately 61%. Moreover, the analysis revealed that the other three functionalized carbohydrates tested, namely, Compounds 6 (24.51 µg/mL), 20 (14.73 µg/mL), and 23 (6.48 µg/mL), also inhibited the growth of the intracellularly deposited tubercle bacilli by 20-30% [13]. In comparison, our tested imidazole thiosemicarbazide derivative caused a 35% decrease in the number of intracellularly growing *Mtb* at a concentration lower (13.48 µg/mL) than that used in the study with carbohydrate derivatives, except for Compound 16A. A more prominent decrease in the number of intracellularly growing *Mtb* of 52% and 69% was also noted in a study with 1H-benzo[d]imidazole derivatives for Compounds 4 (0.749 µg/mL) and 6 (0.037 µg/mL), respectively [35].

The effect of inhibitory concentrations of ITD-**13** on *Mtb* H37Rv biofilm formation was examined by performing biofilm assays in the absence and presence of the derivative (MIC_50_ and MIC_90_) to fully investigate its mycobactericidal potential. After 5 weeks of incubation, the biofilm structure was determined using a Leica SPE confocal microscope (Figure 6).

Macroscopic visualization revealed a clear difference in the biomasses of the compound-treated and untreated biofilms. The formation of biofilms in the presence of ITD-**13** was significantly altered. A tendency for tubercle bacilli cells to form aggregates was observed, with the most significant effect noted at an ITD-**13** concentration corresponding to MIC_90_ (Figure 6A). Furthermore, microscopic analysis of the biofilms confirmed the effect of ITD-**13** on the formation of *Mtb* cell aggregates. Moreover, confocal microscopy images of the biofilm structure showed that the changes in the structure of ITD-**13**-treated biofilms (MIC_50_ and MIC_90_) were related to the decrease in their homogeneity (Figure 6B–D). The observed alterations in *Mtb* biofilms correlated with increasing ITD-**13** concentrations. The effect of the inhibitory concentrations (MIC_50_ and MIC_90_) of ITD-**13** on the viability of biofilm-forming bacilli was also tested by performing live/dead staining (Figure 7). In the presence of ITD-**13**, an increased number of dead bacilli was observed at both tested concentrations of the compound.

The effects of the inhibitory concentrations of ITD-**13** on *Mtb* biofilm formation and on its mature form were also evaluated using REMA. The results showed a significant decrease in the capacity of *Mtb* to form biofilms and in the viability of mature biofilms in the presence of ITD-**13** compared to the untreated control (Figure 8). We observed reductions in the ability of tubercle bacilli to form biofilms in the presence of the derivative of 77.8% at an MIC_50_ concentration (*p* < 0.0001) and 89.4% at an MIC_90_ concentration (*p* < 0.0001) of ITD-**13**. Furthermore, we also noted decreases in the viability of *Mtb* cells forming mature biofilms of 9.4% and 15.5% at the MIC_50_ (*p* < 0.01) and MIC_90_ (*p* < 0.0001) ITD-**13** concentrations, respectively. The reduction in a number of viable tubercle bacilli was directly correlated with the increasing concentration of ITD-**13** and reached 39.3% in the presence of a 2 × MIC_90_ (*p* < 0.0001) concentration of the compound in the culture medium.

*Mtb*, like many other bacteria, form biofilms, which are generally described as a multicellular structure consisting of bacteria and their extracellular polymeric substances (EPS), such as polysaccharides, DNA, and proteins [36]. Despite the lack of surface fimbriae and pili and the lack of synthesis of typical EPS, the formation of mycobacterial biofilms is usually defined similarly to other bacterial biofilms [37]. Mycobacterial biofilms form on surfaces, as well as on the air-media interface [38], which is related to the unique characteristics of the mycobacterial cell wall, in particular the presence of a high lipid content [39]. The development of biofilms in mycobacteria is a complex process regulated by many molecules, including polysaccharides, structural proteins, glycopeptidolipids, GroEL1 chaperones, shorter-chain mycolic acids, and DNA, with the latter two suggested to be crucial for the formation and regulation of *Mtb* biofilms [38,40,41,42]. Furthermore, environmental conditions, such as nutrients, ions, and carbon sources, potentially influence mycobacterial behavior and have a regulatory role in biofilm formation [38,43,44]. Moreover, similar to other bacteria, this process may depend on a number of proteins associated with quorum sensing that ensure intercellular communication [44,45,46]. The ability of bacteria to form biofilms is undoubtedly one of the important pathogenicity factors that makes bacterial pathogens more persistent, more resistant to most antibiotics, and more resilient to host immunity, increasing the difficulty of their eradication [36]. However, the role of biofilms in the pathogenesis of tuberculosis remains to be clarified. The importance of biofilms in this disease is associated with their indicated participation in the process of caseous necrosis and cavitation formation in lung tissue, *Mtb* persistence in the infected host and expanded drug tolerance of *Mtb* strains [19,47].

Studies conducted by Ojha et al. have shown a decrease in the activity of anti-*Mtb* antibiotics against tubercle bacilli biofilms. The experiments revealed that this structure harbors a drug-tolerant mycobacterial population that persists despite exposure to high levels of INH and RMP, two major antibiotics used to treat TB caused by drug-sensitive strains [38]. Another study by Chakraborty et al. also clearly indicated that the formation of biofilms by *Mtb* is relevant to the pathogenesis of tuberculosis and described the contribution of biofilm development to the drug tolerance of tubercle bacilli. The authors confirmed the establishment of *Mtb* biofilms in the lung tissue of infected mice, nonhuman primates and humans. Furthermore, they suggested the substantial relationship between biofilm formation and *Mtb* protection from host immune response mechanisms and the activity of antimycobacterial drugs [44]. The imidazole-thiosemicarbazide derivative ITD-**13**, which was carefully analyzed in our study, possesses the ability to disturb *Mtb* biofilm formation, and this compound also exerts a bactericidal effect on mycobacteria located in a mature biofilm. After considering the results of our study and the importance of the biofilm structure for *Mtb* survival and the infection outcome, ITD-**13** seems to be a very promising compound for further in vivo research aimed at identifying new, effective antimycobacterial drugs.

The acquired resistance of *Mtb* to antituberculosis drugs is typically due to the acquisition of mutations in a gene encoding a drug-target protein or an enzyme activating a particular prodrug within the bacilli. In the present study, the mutation frequency was estimated for acquired tubercle bacilli resistance to ITD-**13** in the presence of 5 × MIC_50_ (67.4 µg/mL) or 5 × MIC_90_ (163 µg/mL), as well as RMP (5 μg/mL) and INH (1 μg/mL) as controls. A fluctuation analysis was performed using the method previously described by Ford et al. [21], and then the drug resistance rate was determined using the Ma-Sarkar-Sandri (MSS) method [22]. The calculated mutation rate was similar for RMP (6.63 × 10^−8^) and ITD-**13** (9.47 × 10^−9^) at both concentrations. Furthermore, the mutation rate observed with INH (8.07 × 10^−6^) that similarly to ITD-**13** possesses hydrazide group [C(=O)NHNH] was significantly higher than that observed for both concentrations of ITD-**13** and RMP (*p* < 0.0001) (Figure 9). Moreover, the in vitro mutation rate correlates well with the bacterial mutation rate in humans, as determined using whole genome sequencing of clinical isolates [21].

We decided to select and analyze mutants resistant to the investigated compound to elucidate the mechanism of action of ITD-**13**. The wild-type *Mtb* strain was cultured in the presence of a subinhibitory concentration of ITD-**13** (0.1 × MIC_90_), and the resistant mutants were selected on solid medium supplemented with 3 × MIC_90_ (97.8 µg/mL) and 5 × MIC_90_ (163 µg/mL) of the compound. We observed that mutant colonies exhibited different morphologies than the wild-type strain of *Mtb* (Figure 10).

Mutants of *Mtb* selected with ITD-**13** compound, as well as the wild-type strain, were subjected to genomic DNA isolation and sequencing using the NGS Illumina system. Bioinformatics analyses identified a number of point mutations in the investigated mutant genomes (Table 2). The analysis revealed that the two genes affected in all investigated mutants were *rv0279c* and *rv0578c*, encoding the nonessential proteins PE_PGRS4 and PE_PGRS7, respectively. However, *rv0279* was the only gene mutated in all mutants at a few different loci. One of six selected mutants carried diverse mutations in *rv3508*, which encodes the nonessential protein PE_PGRS54. Additionally, some selected mutants carried mutations in *rv0197, sigB, rv2813, rv2816c, rv3212,* and *rv3213c* and were not subjected to further investigations.

The identification of mutations in nonessential genes suggests that they are not target sites for the tested compound. On the other hand, the identification of mutations in the genes encoding components of the *Mtb* cell wall may indicate the role of the products (proteins) of these genes in the transport, penetration or binding of the compound. Studies conducted by Korycka-Machała et al. based on the mechanism of anti-*Mtb* activity of *thio*-disaccharides using a similar methodology identified that the PPE51 protein is involved in the uptake and transport of disaccharides by tubercle bacilli [23]. The PPE and PE families possess conserved motifs in their 180- and 110-amino-acid N-terminal regions, Pro-Pro-Glu (PPE) and Pro-Glu (PE), respectively. Furthermore, the PE and PPE protein families are highly polymorphic, localize to the cell surface or are secreted and are expressed during infection, suggesting that they are important factors in TB pathogenesis [48].

The PE-PGRS (proline-glutamic polymorphic GC-rich repetitive sequence) subfamily are PE proteins, whose genes occupy approximately 10% of the *Mtb* genome coding capacity. The *Mtb* genome contains more than 65 *pe_pgrs* genes, which are present in all members of the *Mycobacterium tuberculosis* complex and many other mycobacterial species that cause diseases in humans. PE_PGRS proteins are important in TB pathogenesis [49,50]. PE_PGRS proteins share the same molecular architecture that is characterized by the presence of four main domains: the PE domain, the PGRS, the linker region with the typical GRPLI motif and the unique C-terminal domain [49]. According to recent reports, PE_PGRS proteins participate in the interaction between bacteria and host cells and promote mycobacterial survival by mediating bacterial entry, modulating host antimycobacterial inflammatory responses, lipid metabolism, cell death, and autophagy, and inhibiting macrophage lysosome maturation [50]. Scientific reports indicate that the proteins of the PE_PGRS family mainly constitute a group of mycobacterial cell wall and cell membrane components [51,52,53]. However, these proteins might also be transported and secreted by the ESX-5 secretion system [54,55].

The results obtained by sequencing the DNA isolated from ITD-**13**-selected *Mtb* mutants showed a point mutation inter alia in the *pe_pgrs4* (*rv0279c*) gene. Although the function of the PE_PGRS4 protein is unknown, this protein and PE_PGRS3 are two highly homologous proteins, where the PGRS domain of the two proteins presents an extra GRPLI motif at positions 528–532 and 421–425 for PE_PGRS3 and PE_PGRS4, respectively [49]. The expression of *pe_pgrs3* and *pe_pgrs4* is differentially regulated, with *pe_pgrs4* constitutively expressed and *pe_pgrs3* specifically expressed in response to long-term exposure to low inorganic phosphate concentrations [56].

De Maio et al. described an important role of PE_PGRS3 protein in *Mtb* infection and pathogen survival. They found that the arginine-rich C-terminal domain of PE_PGRS3 is engaged in tubercle bacilli adhesion and entry into host cells, namely, human macrophages and pneumocytes. Furthermore, the authors suggested the involvement of PE_PGRS3 in scavenging lipids, which might be utilized as a carbon source by intracellularly located *Mtb*. Additionally, De Maio et al. documented that the PE_PGRS3 protein, once translocated through the plasma membrane through an ESX5-dependent mechanism [57,58], remains associated with the mycobacterial membrane or may possibly be released to exert its activity and serve as a bridge between pathogen and the target host cells [59].

Treatment of tubercle bacilli with ITD-**13** also induced a mutation in the *Mtb* DNA sequence encoding the *pe_grs7* gene, which encodes the PE_PGRS7 protein. Unfortunately, the function of both the PE_PGRS4 and PE_PGRS7 mycobacterial proteins is unknown and must be investigated. Considering the results obtained and presented in this paper, on the one hand, these proteins may potentially contribute to the proper growth of *Mtb* cells and biofilm formation by this pathogen. On the other hand, ITD-**13**-treated mutants probably did not accumulate mutations in the genes encoding the potential target, and the detected changes in the DNA sequence of the mutants suggest that *Mtb* resistance to ITD-**13** may be phenotypic, observed as morphological changes in pathogen growth and differences in the biofilm structure.

## 4. Conclusions

The obtained results indicate several favorable properties of ITD-**13**, an imidazolo-thiosemicarbazide compound with a 4-diMeaminoPh substitution at the N4 position, such as high antimycobacterial activity accompanied by low cytotoxicity toward host cells, the abilities to inhibit *Mtb* intracellular growth and the formation and maintenance of bacterial biofilms. Considering the importance of intracellular location and biofilm generation in bacterial survival and resistance to both host immunity and chemotherapy, the imidazole-thiosemicarbazide derivative ITD-**13** appears particularly promising as a target for further research aimed at the implementation of new, safe and effective drugs in anti-TB regimens.

## 5. Patents

PL238467B1: 1,4-Disubstituted thiosemicarbazide derivative, method of its preparation and its medical use; applicants: University of Lodz [PL]; Medical University of Lublin [PL]. Inventors: Dzitko Katarzyna [PL]; Dziadek Bożena [PL]; Bekier Adrian [PL]; Kawka Malwina [PL]; Węglińska Lidia [PL]; Paneth Agata [PL]. Classifications IPC A61P31/06; C07D233/70; C07D233/76;

## Figures and Tables

**Figure 1 cells-10-03476-f001:**
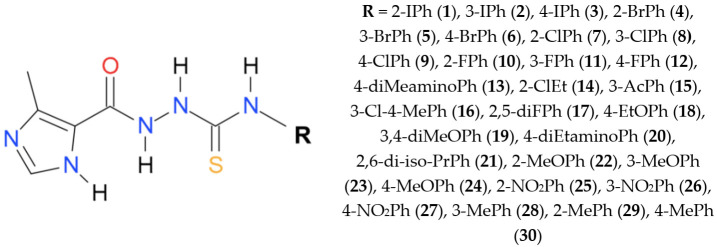
Chemical structure of the imidazole-thiosemicarbazide used in this study.

**Figure 2 cells-10-03476-f002:**
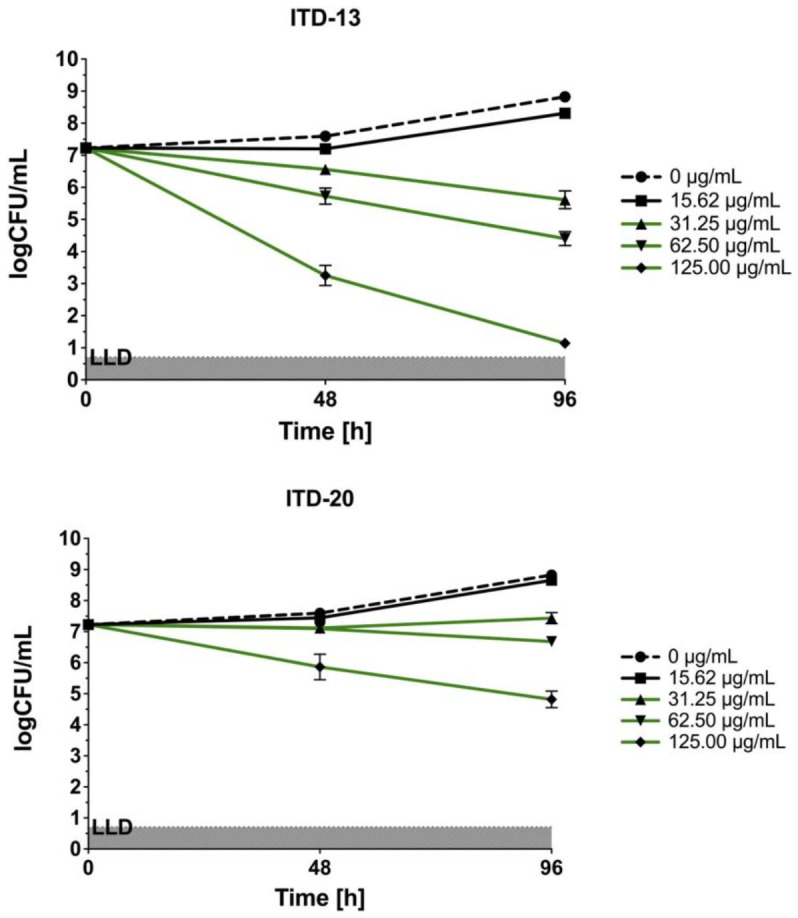
Concentration- and time-dependent bactericidal activities of imidazole-thiosemicarbazide derivatives (ITD-**13**, ITD-**20** and ITD-**30**) against *Mtb* H37Rv. Error bars indicate standard deviations. Curves with statistically significant differences are marked in green (*p* < 0.05). Data were compared using two-way ANOVA followed by Dunnett’s multiple comparisons test. Experiments were performed in triplicate.

**Figure 3 cells-10-03476-f003:**
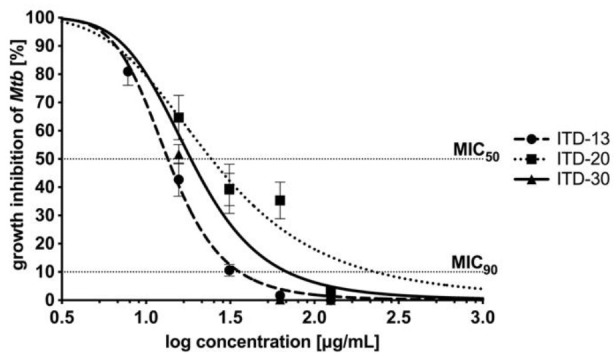
Concentration–effect curves of imidazole-thiosemicarbazide derivatives (ITD-**13**, ITD-**20** and ITD-**30**) against *Mtb* H37Rv after 48 h of incubation. The percentage of growth inhibition was calculated as quotient of the CFU/mL of treated *Mtb* divided by CFU/mL untreated *Mtb*. MIC_50_/MIC_90_ represents the concentrations of tested compounds that were required for 50/90% inhibition of *Mtb* growth. Error bars indicate standard deviations. Experiments were performed in triplicate.

**Figure 4 cells-10-03476-f004:**
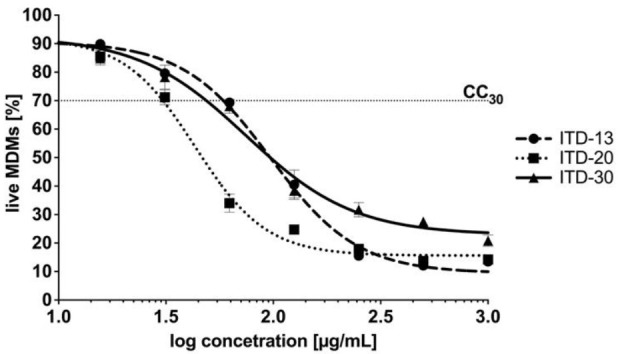
Dose–response curves of imidazole-thiosemicarbazide derivatives ITD-**13**, ITD-**20** and ITD-**30** against MDMs. The following equation was used to calculate the percentage of live MDMs: live MDMs (%) = 100% × [sample A_570_ (the mean value of the measured absorbance of the treated cells)/blank A_570_ (the mean value of the measured absorbance of the untreated cells)]. Cytotoxicity, reported as CC_30_, was defined as the concentration of the tested compound that caused the destruction of ≥30% of macrophages after 48 h of incubation. Error bars indicate standard deviations. Experiments were conducted in triplicate.

**Figure 5 cells-10-03476-f005:**
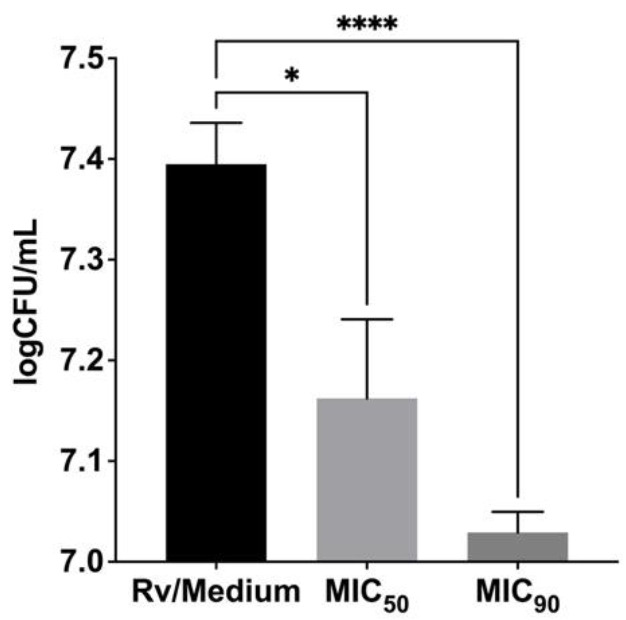
Inhibitory effect of the imidazole-thiosemicarbazide derivative ITD-**13** on intracellularly growing *Mtb* H37Rv in human monocyte-derived macrophages. The number of intracellularly located live bacteria was calculated by determining CFUs, and viability was reported as logCFU/mL. Error bars indicate standard deviations. Values with statistically significant differences are labeled by brackets and asterisks as follows: *, *p* < 0.05; ****, *p* < 0.0001. Data were analyzed using the Mann–Whitney U-test to compare the anti-*Mtb* activity of ITD-**13**.

**Figure 6 cells-10-03476-f006:**
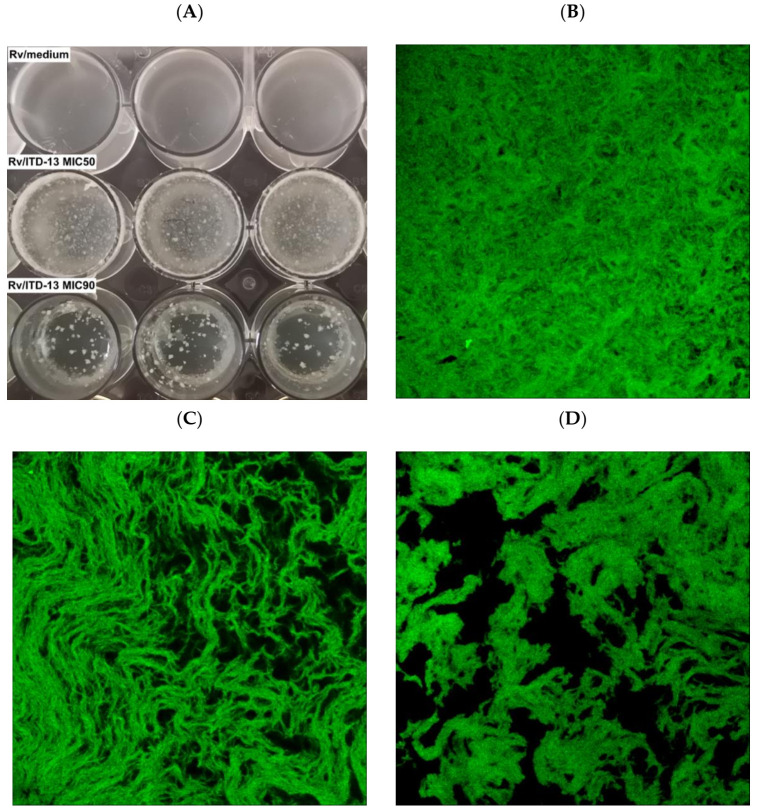
Macroscopic (**A**) and confocal microscopy images of the biofilm structures not treated with ITD-**13** (**B**), treated with the MIC_50_ (**C**) and treated with the MIC_90_ (**D**).

**Figure 7 cells-10-03476-f007:**
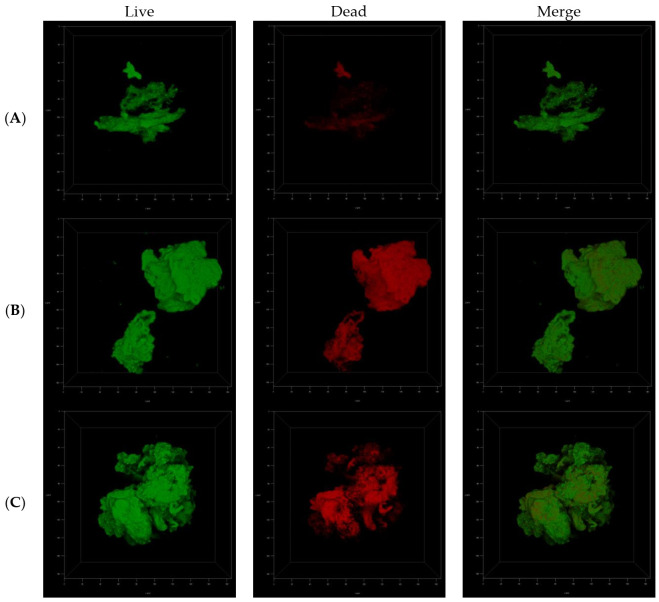
Confocal microscopy images of live/dead cells in the three-dimensional biofilm structure not treated with ITD-**13** (**A**), treated with an MIC_50_ concentration of ITD-**13** (**B**) and treated with an MIC_90_ concentration of ITD-**13** (**C**).

**Figure 8 cells-10-03476-f008:**
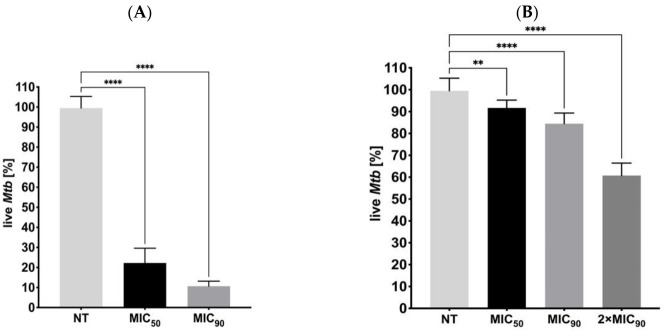
Effects of ITD-**13** on *Mtb* biofilm formation (**A**) and on mature biofilms (**B**). Error bars indicate standard deviations. Values with statistically significant differences are labeled by brackets and asterisks as follows: **, *p* < 0.01; ****, *p* < 0.0001. Data were compared using two-way ANOVA followed by Dunnett’s multiple comparisons test.

**Figure 9 cells-10-03476-f009:**
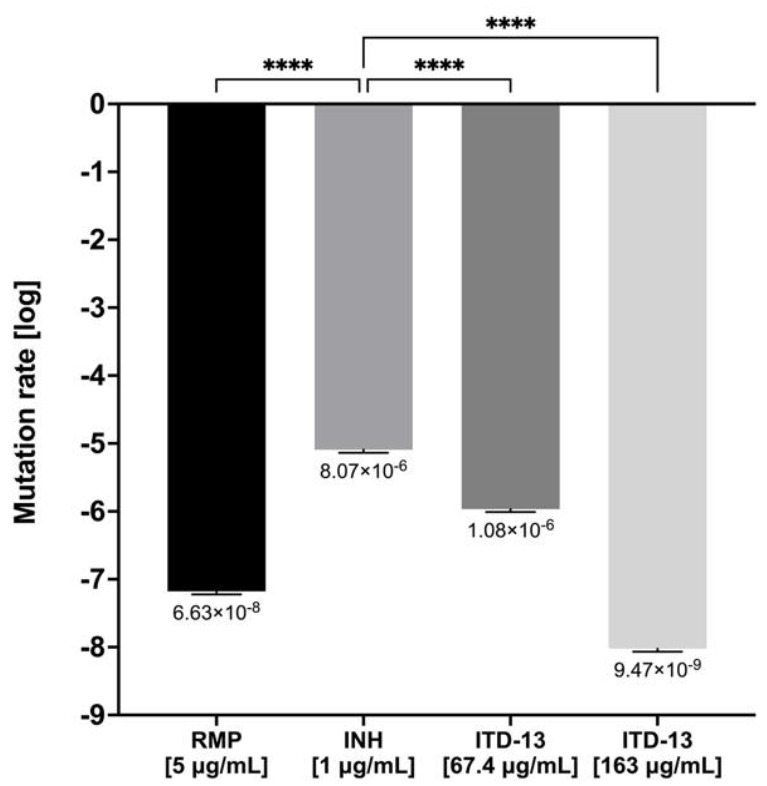
The drug resistance rate calculated for ITD-13 and controls (RMP and INH). Error bars indicate standard deviations. Values with statistically significant differences are labeled by brackets and asterisks as follows: ****, *p* < 0.0001. Data were compared using two-way ANOVA followed by Dunnett’s multiple comparisons test. RMP—rifampicin; INH—isoniazid.

**Figure 10 cells-10-03476-f010:**
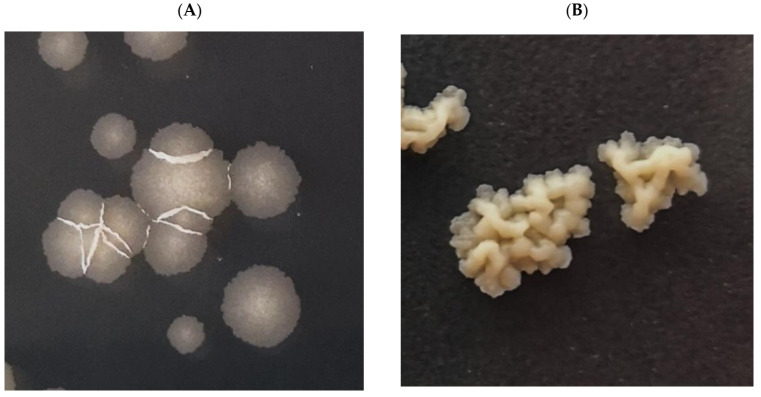
Macroscopic images of colonies of the *Mtb* H37Rv strain growing on 7H10/OADC/Tween/glycerol plates supplemented with or without ITD-**13**. (**A**) Wild-type *Mtb* H37Rv and (**B**) mutants selected on solid medium supplemented with compound ITD-**13**.

**Table 1 cells-10-03476-t001:** MICs of imidazole-thiosemicarbazide derivatives against *Mtb* H37Rv obtained using REMA.

MIC [μg/mL]	Compounds
125.00	ITD-**1**, ITD-**2**, ITD-**4**, ITD-**15**, ITD-**21**, ITD-**25**
62.50	ITD-**3**, ITD-**5**, ITD-**6**, ITD-**7**, ITD-**8**, ITD-**9**, ITD-**10**, ITD-**11**, ITD-**12**, ITD-**16**, ITD-**17**, ITD-**18**, ITD-**19**, ITD-**22**, ITD-**23**, ITD-**24**, ITD-**26**, ITD-**27**, ITD-**28**, ITD-**29**
31.25	ITD-**20**, ITD-**30**
15.62	ITD-**13**
*NA*	ITD-**14**

ITD—imidazole-thiosemicarbazide derivatives; NA—not active against *Mtb*.

**Table 2 cells-10-03476-t002:** Detection of SNP variants in sequencing data derived from mutagenized strains was performed using the breseq algorithm. The results of variant calling for each sample were filtered against the H37Rv-type strain used for mutagenesis.

Position	Mutation	Rv_M1	Rv_M3	Rv_M4	Rv_M5	Rv_M7	Rv_M8	Annotation	Gene
234,496	+G				+		+	coding (2266/2289 nt)	*rv0197*
336,701	A→G	+	+	+	+	+	+	G791G (GGT→GGC)	*rv0279c*
338,100	T→C	+	+		+			N325S (AAC→AGC)	*rv0279c*
338,453	A→G	+	+	+		+	+	A207A (GCT→GCC)	*rv0279c*
672,491	C→G	+	+	+	+	+	+	G1142G (GGG→GGC)	*rv0578c*
3,022,903	A→G	+		+	+	+	+	Q148R (CAG→CGG)	*sigB*
3,123,354	Δ72 bp				+			intergenic (+4318/+200)	*rv2813*/r*v2816c*
3,590,686	+C	+	+	+	+	+	+	intergenic (+69/+6)	*rv3212*/r*v3213c*
3,935,494	C→A		+					A1497D (GCC→GAC)	*rv3508*

*rv0197*—oxidoreductase; *rv0279c*—PE-PGRS family protein PE_PGRS4; *rv0578c*—PE-PGRS family protein PE_PGRS7; *rv3508*—PE-PGRS family protein PE_PGRS54; *sigB*—RNA polymerase sigma factor SigB; *rv2813/rv3212*—hypothetical protein; *rv2816c*—CRISPR-associated endoribonuclease Cas2; *rv3213c*—ParA-like protein. +: presence of indicated mutation. The mutated nucleotides are indicated by red fonts.

## Data Availability

Data are contained within the article.

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
