# Peer review of "Imidazole-Thiosemicarbazide Derivatives as Potent Anti-Mycobacterium tuberculosis Compounds with Antibiofilm Activity"

_cells, 2021, doi:10.3390/cells10123476_

Round 1

Reviewer 1 Report

Bekier and the group have studied the in vitro activity of new 23 imidazole-thiosemicarbazide derivatives (ITDs) against Mtb growth and biofilm formation. Furthermore, the authors evaluated the cytotoxic ability of these compounds against L929 cells and human monocyte-derived macrophages (MDMs). Mutation frequency and whole-genome sequencing of mutants that were resistant to ITDs were also performed. The antitubercular activity of ITDs was tested against macrophages and they were found to inhibit tubercular bacilli growth. The study is interesting and can be accepted for publication.

Comments

  1. Authors should compare these imidazole TSC analogs with INH as they are structurally similar.
  2. This chemical class of compounds is prone to hydrolysis. I suggest authors conduct hydrolytic stability studies to evaluate the stability in aqueous conditions.
  3. What is the water solubility of these compounds?
  4. INH has been shown to lose its antiTB activity in low iron conditions resulting in INH resistance. TB patients tend to show lower iron levels clinically. I recommend authors study the anti TB activity in low iron conditions which will also help these compounds to treat INH resistant MTb cases.
  5. Further mechanistic studies are needed to find out the exact mechanism of actions.
  6. The structure of the TSC is also reported to be an iron chelator, something to think about in terms of mechanism of action.

Author Response

Comments from the Editors and Reviewers:

Reviewer: 1

Bekier and the group have studied the in vitro activity of new 23 imidazole-thiosemicarbazide derivatives (ITDs) against Mtb growth and biofilm formation. Furthermore, the authors evaluated the cytotoxic ability of these compounds against L929 cells and human monocyte-derived macrophages (MDMs). Mutation frequency and whole-genome sequencing of mutants that were resistant to ITDs were also performed. The antitubercular activity of ITDs was tested against macrophages and they were found to inhibit tubercular bacilli growth. The study is interesting and can be accepted for publication.

1. Authors should compare these imidazole TSC analogs with INH as they are structurally similar.

Appropriate sentence has been added to the manuscript.

2. This chemical class of compounds is prone to hydrolysis. I suggest authors conduct hydrolytic stability studies to evaluate the stability in aqueous conditions.

Since we have previously evaluated the activity of imidazole-thiosemicarbazide derivatives against the parasite Toxoplasma gondii [11,12], their hydrolytic stability at physiological pH was also checked before biological studies have been performed. It was found that all tested compounds were stable in phosphate buffer at pH = 7.4 and neither decomposition nor dehydrocyclization to 1,2,4-triazole or 1,3,4-thiadiazole system was observed (Heteroatom Chemistry 2010, 21, 521-525). 

3. What is the water solubility of these compounds?

Water solubility of ITD-13 is to 125 mg/L in 25 °C but in dimethyl sulfoxide (DMSO) more than 10×104 mg/L in 25 °C. If our derivatives were firstly dissolve in DMSO to 50 mg/mL and then dilute in medium to 1 mg/mL were still soluble. In this approach, we increased eightfold water solubility of our derivatives from 125 mg/L to 1000 mg/L.

4. INH has been shown to lose its antiTB activity in low iron conditions resulting in INH resistance. TB patients tend to show lower iron levels clinically. I recommend authors study the anti TB activity in low iron conditions which will also help these compounds to treat INH resistant MTb cases.

INH presents bactericidal effect against metabolically active Mtb but not latent bacteria. It was reported that the MIC for INH increases from 0.125 to 32 ug/ml if transferring bacilli from high to low iron conditions (doi: 10.1016/j.bmcl.2014.09.032). In vitro, INH rapidly kills 99.0% to 99.9% of INH-susceptible M. tuberculosis bacilli during the first 3 days of treatment. The persisters (0.1-1 %) are INH-susceptible, non- or slowly-replicating, and adapting to stress in order to survive (doi: 10.1016/j.jmb.2019.02.016.).The phenotypic resistance to INH was linked in a number of papers to growth rate, nutrient starvation, anaerobic conditions, chronic infection in mice and a number of mechanisms were proposed to explain the process including stringent response, two-component system MprAB, respiration. We agree with Referee that it would be useful to test whether the most promising imidazole-thiosemicarbazide derivatives are active against bacilli in latent stage. In this story we estimated the MIC of the compounds against actively growing bacilli, however, we also observed the activity of the tested compounds against biofilm composing bacteria which are known to be highly differentiated and often phenotypically resistant for treatment with antibiotics. More studies are required to determine the mycobactericidal effect of the compounds in a various conditions including nutrient starvation, anaerobic and low iron conditions.

5. Further mechanistic studies are needed to find out the exact mechanism of actions.

We are still working on a specific molecular target for imidazole TSC in Mtb.

6. The structure of the TSC is also reported to be an iron chelator, something to think about in terms of mechanism of action.

Thank you for this interesting suggestion. We will focus on this issue in the next experiments.

Reviewer 2 Report

1. The scope of this study does not seem to fit the Cells journal. It would be appropriate to submit it to another journal.
2. A detailed description of the experimental method for the synthesis of Imidazole-thiosemicarbazide derivatives is required.

Author Response

Response to comments

Detailed responses (marked in blue) to reviewers’ comments

Reviewer: 2

1. The scope of this study does not seem to fit the Cells journal. It would be appropriate to submit it to another journal.

Thank you for this suggestion, but our other previous research, similar to this one, was successfully published in the Cells:

  • PPE51 Is Involved in the Uptake of Disaccharides by Mycobacterium tuberculosis https://doi.org/10.3390/cells9030603
  • 1,3,4-Thiadiazoles Effectively Inhibit Proliferation of Toxoplasma gondii https://doi.org/10.3390/cells10051053

2. A detailed description of the experimental method for the synthesis of Imidazole-thiosemicarbazide derivatives is required.

A detailed description of the experimental method for the synthesis of Imidazole-thiosemicarbazide derivatives has been added as suggested.

Reviewer 3 Report

In this study, Bekier et al. describe the use of new imidazole-thiosemicarbazide derivatives (ITDs) against Mtb infection and their effects on mycobacterial biofilm formation. Authors identify a highly potential anti-tb reagent (ITD-13) through MIC assay, time-kill curves, bacterial intracellular growth and the effect on biofilm formation. They also found several mutants that were resistant to ITDs.  Overall, I think this is a comprehensive and informative study that would be a great addition to the cells.

Author Response

Thank you for your review.

Round 2

Reviewer 2 Report

My concerns has been addressed.